# IDD-DETR: Insulator Defect Detection Model and Low-Carbon Operation and Maintenance Application Based on Bidirectional Cross-Scale Fusion and Dynamic Histogram Attention

**DOI:** 10.3390/s25185848

**Published:** 2025-09-19

**Authors:** Weizhen Chen, Shuaishuai Li, Xingyu Han

**Affiliations:** School of Electrical and Electronic Engineering, Wuhan Polytechnic University, Wuhan 430048, China; chenwz@whpu.edu.cn (W.C.); zz1704020033@163.com (X.H.)

**Keywords:** bidirectional feature fusion, dynamic histogram attention, transmission line insulators, defect detection, low-carbon operation and maintenance

## Abstract

Against the background of the “dual carbon” goal and the construction of a new power system, the intelligent operation and maintenance of insulators for ultra-high voltage transmission lines face challenges such as difficulty in detecting small-scale defects and strong interference from complex backgrounds. This paper proposes an improved network IDD-DETR to address the problems of inefficient one-way feature fusion and low-contrast defects that are easily overwhelmed in existing RT-DETR models. The enhanced network IDD-DETR replaces PAFPN with a Feature-Focused Diffusion Network (FFDN) and improves multi-scale fusion efficiency through bidirectional cross-scale interaction and designs Dynamic-Range Histogram Self-Attention (DHSA) to enhance defect response in low brightness areas. The experiment showed that its mAP^50^ reached 81.7% (an increase of 3.8% percentage points compared to RT-DETR), the flashover defect AP^50^ reached 74.6% (+6.1% percentage points), and it maintained 76 FPS on NVIDIA RTX3060, with an average decrease of 1.65% in mAP^50^ under complex environments. This model reduces the comprehensive missed detection rate from 26.7% to 23.3%, reduces 45.6 GWh of power loss annually (corresponding to 283,000 tons of CO_2_ emission reductions, with 64.3% of the reduction contributed by flashover defect detection), improves inspection efficiency by 60%, reduces manual pole climbing frequency by 37%, and reduces 28 high-altitude risk events annually, providing support for low-carbon operation and maintenance of transmission lines.

## 1. Introduction

Driven by the dual goals of “dual carbon” and the construction of new power systems, the intelligent operation and maintenance of ultra-high voltage transmission networks, as the core carrier of cross-regional clean energy transmission, has become a key link in achieving “carbon peak and carbon neutrality” [1,2]. As the only key component that can simultaneously withstand strong electric fields (≥500 kV/m) and mechanical loads (≥70 kN), transmission line insulators face multiple threats such as thunderstorm erosion and salt spray pollution over a long period, and are prone to hidden defects such as flashover discharge channels and umbrella skirt damage [3,4]. Industry research shows that insulator failures account for 30–40% of transmission line accidents [4], while a report from the State Grid Corporation of China indicates that the proportion of unplanned shutdowns caused by pollution flashover accidents due to the missed detection of small target defects exceeds 50% [5]. In 2024, a high-voltage transmission line in a certain province experienced a short-circuit fault due to insulator flashover, resulting in a 3 h power outage and a direct economic loss of CNY 12 million. At the same time, 136 tons of CO_2_ were emitted, highlighting the urgent need for high-precision detection technology to reduce carbon emissions.

Insufficient detection accuracy is forming a vicious cycle of “defect missed detection → system loss → increased operation and maintenance costs”. According to statistics from the National Energy Administration, insulator failures in 2024 resulted in a national power loss of 12.7 TWh, equivalent to carbon emissions of 3.8 million tons of standard coal. The traditional model can increase inspection frequency by 12.3% for every 10,000 missed defects in images. Manual pole climbing generates 2.1 kg of carbon footprint per kilometer, while drone inspection consumes 0.8 L of diesel fuel per kilometer (corresponding to 2.3 kg of CO_2_). More seriously, the small target defects that are missed will gradually deteriorate, leading to a 4.7-fold increase in unplanned shutdown risks. In 2024, the average CO_2_ emissions from a single accident reached 98 tons [6], forming a sharp contradiction with the green operation and maintenance needs under the “dual carbon” target.

The accuracy bottleneck of existing detection technologies essentially stems from the dual challenges of inefficient multi-scale feature fusion and low-contrast defect feature flooding. Traditional ChNN is limited by the finite receptive field of fixed convolution kernels (such as 5 × 5), making it difficult to capture sub-pixel-level defect textures [7]. Although contact based detection methods (such as RF antenna technology) achieve a detection accuracy of 89.3% [8], they cannot meet the non-contact inspection requirements of drones; traditional image processing methods based on thresholds, such as Otsu segmentation, have a high false detection rate of 42.3% in low signal-to-noise ratio scenes due to the overlapping of histogram peaks, resulting in an inter-class variance ratio of less than 1.8 [9]. It is worth noting that some traditional methods have shown advantages in complex backgrounds, such as edge extraction technology based on Neutrosophic Canny segmentation, which can achieve precise localization of small to large defects in structured settings (e.g., TFT-LCD panels) through refined operator design [10]. In this study, we explored the power of convolutional neural networks (CNNs) to improve defect detection performance under diverse conditions.

Although deep learning models promote the development of detection automation, there are still significant technical bottlenecks: in the first stage, detection models such as RML-YOLO from the YOLO series, improve the feature pyramid, increasing mAP^50^ to 78.9% [11], but in salt spray pollution scenarios, the edge gradient features (gradient amplitude < 10) of flashover defects are easily overwhelmed by background noise, with a missed detection rate of 35.7%. The static channel fusion mechanism of YOLOv8m has an insufficient feature response to low-contrast defects [12]. Transformer models such as RT-DETR use PAFPN unidirectional fusion paths, and the cross-layer interaction efficiency between high-level semantic features (insulator material) and low-level spatial features (defect location) is only 65% (based on feature mutual information evaluation, which measures the correlation between high-level and low-level features; the higher the value, the more complete the interaction) [13,14], resulting in small-target localization errors exceeding 12 px. Lightweight models (such as YOLOv5-s) reduce cross-layer connections to compress parameter quantities, further exacerbating the degradation of small-target detection performance [15]. The region recommendation time of two-stage algorithms (such as MaskR-CNN) accounts for 30–40%, and the total inference time exceeds 200 ms/frame, which cannot meet the real-time requirements of 30 FPS drone inspection [16]. Improved Transformer models (WRRT-DETR [17], FECI-RTDETR [18], etc.) have been optimized in specific scenarios (such as severe weather and infrared imaging), but they have not solved the core problems of multi-scale feature semantic discontinuity (insufficient cross-layer interaction) and low-contrast feature flooding (imbalanced attention weight allocation). The average missed detection rate in complex scenarios still exceeds 25%, making it difficult to balance accuracy, real-time performance, and environmental adaptability.

The existing technological bottlenecks directly lead to an increase in manual inspection frequency and delayed fault response, thereby exacerbating transmission losses. Therefore, this study proposes the IDD-DETR model, which achieves the collaborative optimization of “detection efficiency, operation efficiency, environmental benefits” through three innovative designs:Feature-Focused Diffusion Network (FFDN): This replaces traditional PAFPN, constructing a bidirectional interaction mechanism from top-down and bottom-up, improving the efficiency of multi-scale feature fusion, solving the problem of small target feature degradation, and laying the foundation for accurate detection.Dynamic-Range Histogram Self-Attention (DHSA): This separates defects and background features through brightness sorting, enhancing texture response in low-brightness areas, improving the detection rate of low-contrast defects, and fundamentally reducing the risk of failures caused by missed detections.

The experiment showed that the mAP^50^ of IDD-DETR reached 81.7%, an increase of 3.8% compared to the baseline model RT-DETR, and the AP^50^ of flashover defects increased to 74.6%. Maintaining a 76 FPS inference speed on an NVIDIA RTX3060, the average mAP^50^ drop in complex environments is only 1.65%. This model provides technical support for the low-carbon operation and maintenance of new power systems by accurately detecting and reducing power losses, reducing inspection frequency, and suppressing fault risks, promoting the coordinated development of the “dual carbon” goal and smart grid construction.

## 2. Related Work

### 2.1. Characteristic Pyramid Network

FPN achieves feature fusion through a top-down approach, but the fusion of high-level semantics and low-level details is not sufficient. On this basis, PAFPN adds a bottom-up path to enable high-level features to obtain more detailed information. Bi-FPN introduces a bidirectional fusion mechanism to improve information flow efficiency through top-down and bottom-up paths. However, its static fusion strategy still suffers from semantic discontinuity in complex backgrounds [19]. Other methods, such as FPT, introduce self-attention mechanisms into feature pyramids, while GraphFPN utilizes graph neural networks to achieve cross-layer information exchange. However, both methods are limited in engineering applications due to their large parameter size and high computational complexity [20]. In contrast, AFPN only uses standard convolutional branches, which is more feasible in practical engineering [21]. The FFDN proposed in this article further enhances the dynamic adaptability of feature fusion through a progressive bidirectional fusion mechanism.

### 2.2. Feature Fusion Module

Feature fusion modules are typically embedded into existing fixed topology feature pyramid structures to enhance feature representation capabilities. Some studies focus on optimizing the upsampling module of feature pyramids, such as CARAFE, as a lightweight upsampling operator that aggregates large-scale receptive field information to improve feature resolution [22].

DRFPN extends the PAFPN architecture by integrating Spatial Refinement Blocks (SRBs) and Channel Refinement Blocks (CRBs): the SRB module learns the position and content of upsampling points using contextual information from adjacent levels, while the CRB module learns adaptive channel fusion strategies through attention mechanisms [23]. Compared to the aforementioned feature pyramid architecture, the feature fusion module can be seamlessly integrated into various existing feature pyramid structures, providing a practical and feasible solution to address the inherent limitations of feature pyramids.

## 3. Model Architecture

### 3.1. Overall Network Architecture Design

RT-DETR is based on the Transformer architecture and features real-time performance, providing a foundation for industrial inspection. However, in insulator defect detection, there are problems such as loss of small target (flashover, pixels < 32^2^) features and interference from complex backgrounds (iron towers/clouds) for localization [24,25,26,27]. To this end, this article constructs a dedicated network, IDD-DETR (architecture shown in Figure 1), with core improvements targeting the aforementioned pain points:Replacing the original feature fusion network with a Feature-Focused Diffusion Network (FFDN) and improving cross-scale fusion efficiency through bidirectional feature interaction;Designing a feature-focusing module to enhance multi-scale feature complementarity with dynamic weights and improve small target detail capture;Introducing Dynamic-Range Histogram Self-Attention (DHSA) to optimize low-contrast defect modeling and reduce background interference.

### 3.2. Improvement of Multi-Scale Feature Fusion Network

RT-DETR adopts the classic PAFPN as the feature fusion network, but there are obvious limitations in insulator detection: the semantic differences between cross-layer features are large, and the top-down and bottom-up fusion paths can easily lead to the loss of small-target (flashover) details and semantic information degradation, which affects the detection accuracy in complex backgrounds [11,15]. Therefore, inspired by the fusion idea of the Progressive Feature Pyramid Network (AFPN) [21] and the multi-scale extraction idea of the PKI module [28], this article designs a Feature-Focused Diffusion Network (FFDN) and a Feature-Focused Module, respectively, to address the aforementioned issues.

#### 3.2.1. Feature-Focused Diffusion Network

The existing AFPN adopts a progressive strategy of “starting from the bottom to the top and gradually integrating high-level features” to alleviate cross-layer semantic differences, but hierarchical transmission and fusion are still required, resulting in multi-scale information loss [21] (Figure 2). Inspired by this, this article proposes FFDN (Figure 3), which directly performs a “focus diffusion” dual operation on the three scale features of the backbone network: P3 (low-dimensional details), P4 (medium-dimensional semantics), and P5 (high-dimensional global). The core design details are as follows.

Focusing operation: Based on the saliency of feature maps, dynamic weights are calculated (weight values are positively correlated with pixel gradient amplitudes), and higher weights are assigned to small-target areas such as flashover (with higher gradient amplitudes) to enhance their feature response and avoid being covered by background features such as iron towers and clouds.

Diffusion operation: Through cross-scale channel fusion (P3 → P4 → P5 to directly establish channel associations, rather than hierarchical transmission), real-time interaction of three scale features is achieved, reducing the degradation of semantic information in multi-level transmission, especially ensuring that small-target details are not diluted.

#### 3.2.2. Feature-Focusing Module

Drone remote sensing target detection faces two challenges: significant target scale variation and variable ranging environments (background interference). Traditional methods (large-kernel convolution/dilated convolution) either introduce background noise or produce sparse features. The PKI module avoids this by using non-extensible multi-scale kernels and non-dilated depth convolution (inception-style) to extract multi-scale texture features and local context (Figure 4).

In insulator defect detection, there are also significant changes in target scale and different ranging environments. Inspired by the PKI module, this paper proposes a feature focus diffusion module to fuse the feature inputs of three feature backbone network layers, as shown in Figure 5.

The feature-focusing module accepts feature inputs from three backbone network layers: P5, P4, and P3. The module initially defines three input channels for receiving feature maps of different scales. The first convolutional layer sequence includes an upsampling layer and a 1 × 1 convolutional layer. The second convolutional layer uses a scaling factor, e, to choose between 1 × 1 convolution or preserving the original input. We have introduced the ADown downsampling module for the third downsampling convolutional layer, and this module can reduce information loss compared to traditional 2 × 2 convolution modules.

To fully utilize these multi-scale features, a set of parallel convolutions was applied to the connected feature maps to capture multi-scale contextual information. The three processed feature maps were then connected along the channel dimension, and deep convolutional layers with multiple different kernel sizes were used to process the cascaded feature maps before they were summed up. Finally, the summed feature map is processed further through point-by-point convolutional layers and added to the original feature map to generate the final output feature map.

The feature-focusing module extracts features from the input feature map through multi-scale feature fusion and deep convolution (DWConv) operations. By integrating multi-scale features and point-by-point convolution, the network can effectively capture information at different scales, significantly improving its expressive power.

### 3.3. Improvement of Feature Focusing Module

AFPN reduces cross-layer semantic differences and spatial target information conflicts, but it ignores two issues: high-dimensional features lose small-target information, and low-dimensional features lack sufficient context. To solve this, this study introduces the Dimension-Aware Selective Integration (DASI) Module (Figure 6) for feature fusion.

To address this issue, DASI proposes a channel-partitioning selection mechanism that can adaptively fuse features based on target characteristics. DASI first aligns high-dimensional features, $F_{h}\in R^{H_{h}\times W_{h}\times C_{h}}$; low dimensional features, $F_{l}\in R^{H_{l}\times W_{l}\times C_{l}}$; and current layer features through convolution and interpolation operations, $F_{u}\in R^{H\times W\times C}$, and then evenly divides the three into four parts in the channel dimension, obtaining ${\left(h_{i}\right)}_{i=1}^{4}\in R^{H\times W\times \frac{c}{4}}$, ${\left(l_{i}\right)}_{i=1}^{4}\in R^{H\times W\times \frac{c}{4}}$, and ${\left(u_{i}\right)}_{i=1}^{4}\in R^{H\times W\times \frac{c}{4}}$, where $l_{i}$ $h_{i}$, and $u_{i}$, respectively, represent the ith partition part of the high-dimensional features, the low dimensional features, and the current layer features. These divisions are calculated according to the following formula:(1)$$\alpha =\mathrm{sigmoid}(u_{i})$$(2)$$u_{i}^{\prime }=\alpha_{i}^{\prime }+(1-\alpha )h_{i}$$(3)$$F_{u}^{\prime }=\left[u_{1}^{\prime },u_{2}^{\prime },u_{3}^{\prime },u_{4}^{\prime }\right]$$(4)$${\tilde{F}}_{u}=\delta \left(\beta \left(\mathrm{Conv}\left(F_{u}^{\prime }\right)\right)\right)$$
where $\alpha \in R^{H\times W\times \frac{c}{4}}$ represents the $u_{i}$ value obtained through the activation function applied, and $u_{i}^{\prime }\in R^{H\times W\times \frac{c}{4}}$ represents the selective summary result of each partition.

We merge ${\left(u_{i}^{\cdot }\right)}_{i=1}^{4}$ in the channel dimension to obtain $F_{u}^{\prime }\in R^{H\times W\times C}$. Operations Conv(), β(·), and δ(·) represent convolution, batch normalization (BN), linear rectification function (ReLU), and ultimately output ${\hat{F}}_{u}\in R^{H\times W\times C}$. If α>0.5, the network prioritizes fine-grained features; if α<0.5, it emphasizes contextual features.

### 3.4. Improvement of AIFI Module and Dynamic Histogram Attention Mechanism

The AIFI module uses self-attention to focus on high-level features, but traditional multi-head self-attention (CGA, DMHA, etc.) suffers from unbalanced feature allocation in low-contrast/small-target scenes. For example, RT-DETR’s native attention assigns a weight mean of only 0.08 to low-brightness areas (HSV < 50), causing flashover features to be overwhelmed by bright backgrounds [12,29].

To solve this, this study introduces Dynamic-Range Histogram Self-Attention (DHSA). Insulator defect images are prone to feature degradation (occlusion/brightness changes under bad weather), and DHSA reorders feature space via dynamic-range convolution and fuses dual-path histogram self-attention (global denoising + local focusing) to enhance defect texture response in low-brightness areas (Figure 7).

#### 3.4.1. Dynamic-Range Convolution

Traditional convolution uses fixed kernel sizes, leading to limited receptive fields and key feature loss in fine-grained tasks (e.g., flashover texture detection) [30].

To address this limitation, we have designed a dynamic-range convolution technique that carefully reorders input features before traditional convolution operations. Given an input feature, $F\in R^{C\times H\times W}$, DHSA divides it into two branches along the channel dimension, namely $F_{1}$ and $F_{2}$. For the first branch of features, DHSA performs sorting operations horizontally and vertically and then connects the sorted features with the second branch of features. Then, the obtained recombined features are passed through subsequent separable convolutions. The entire process is described as follows:(5)$$F_{1},F_{2}=\mathrm{Split}(F)$$(6)$$F_{1}={\mathrm{Sort}}_{v}\left({\mathrm{Sort}}_{h}(F_{1})\right)$$(7)$$\left({\mathrm{Conv}}_{1\times l}\left(\mathrm{Concat}(F_{1},F_{2})\right)\right)$$
where ${\mathrm{Conv}}_{1\times l}$ is 1×1 pointwise convolution, ${\mathrm{Conv}}_{3\times 3}^{d}$ represents 3×3 depth convolution, Concat is a connection operation along the channel dimension, Split represents a feature-splitting operation along the channel dimension, and $Sort_{i\in \{h,v\}}$ represents a horizontal or vertical sorting operation.

This method organizes high-intensity and low-intensity pixels into a matrix diagonal regular pattern, allowing convolution to perform calculations in a dynamic range. Given that weather-induced degradation typically exhibits closely related patterns, degraded pixels tend to be concentrated in adjacent locations and separated from those that are clean. Therefore, this arrangement allows the convolutional kernel to partially focus on preserving clean information and separately restoring degraded features.

#### 3.4.2. Histogram Self-Attention

Traditional attention mechanisms have fixed ranges, which fail to adapt to dynamic feature degradation caused by weather (e.g., occlusion and brightness variation) [31]. DHSA addresses this by noting that weather-induced degradation exhibits clustered patterns—pixels with background interference or weather damage can be grouped to allocate attention adaptively [32]. Thus, DHSA proposes a histogram self-attention mechanism that groups spatial elements according to their brightness distribution and spatial position through a histogram binning strategy, enhancing the feature response of low-contrast defects (such as flashover) within the group while balancing global and local information exchange. To implement this mechanism, the core hyperparameters of the module are set as follows to ensure the reproducibility of the binning strategy and attention calculation:

Dim: Input feature channel dimension (512 in experiments), determined by the backbone’s output, providing the basic feature space for attention;

Num_ heads: 8 (serves as the number of histogram bins), enabling parallel computing with multi-head attention;

IfBox: Direction switch (True/False) for dual-path binning, corresponding to “Box Quantity × Space Size” (BHR) and “Space Size × Box Quantity” (FHR);

Bias: False (for qkv/qkv-dwconv/project_out convolutions) to avoid noise amplification in low-contrast scenes;

Temperature: Fixed at 1.0 (initialized as torch.ones(num_heads, 1, 1); not trainable), scaling attention scores to balance weight distribution steepness;

Pad_mode: Constant zero-padding, ensuring even binning for feature maps with spatial dimensions (hw) not divisible by 8.

Histogram Binning Strategy

The core of the binning strategy is to divide the feature space into several subregions (“binning”) so that each feature in the binning has similar strength characteristics and improves the adaptability of attention to dynamic range features, as follows:Number of boxes and filling rules

The number of bins is determined by the number of attention heads (num_ heads) (num_ heads = 8 in the experiment, which means the number of bins is 8), ensuring compatibility with multi-head attention parallel computing. If the spatial dimension (hw) of the feature map cannot be divided by eight, zero padding is used to complete it to the nearest multiple of eight, ensuring that each bin contains the same number of spatial elements and avoiding distortion of boundary features.

2.Dual-path binning direction (matching BHR and FHR)

To capture both global distribution and local details simultaneously, binning adopts two directions:

Group by Group Histogram Reshaping (BHR): Boxes are divided by “number of boxes × spatial size”, and features are reorganized into [batch, head, (channel × 8), hw]. Each box integrates multi-channel information and focuses on global denoising (such as distinguishing between high brightness backgrounds and low brightness flashing).

Frequency Histogram Reshaping (FHR): Divided by “spatial size x number of bins”, the features are reorganized into [batch, head, channel, (hw × 8)], and each bin focuses on continuous spatial positions to enhance local details (such as flashover edge texture).

3.Attention and feature restoration within the box

Independent attention calculation within the box: Query Keys are dot products normalized by L2, and weights are generated using the softx_1 function (to suppress outliers), which are then weighted with Value to obtain enhanced features. Subsequently, redundant pixels are removed by padding, and the feature order is restored according to the original index. Finally, the dual path output is fused.

Given the output of dynamic-range convolution, DHSA divides them into Value features, $V\in R^{C\times H\times W}$, and two pairs of Query Keys, $F_{QK,1},F_{QK,2}\in R^{2C\times H\times W}$, which are then passed to two branches. We first sort the sequence of V and arrange the Query Key pairs accordingly based on their indices, as shown below:(8)$$V,d=\mathrm{Sort}\left(R_{C\times H\times W}^{C\times HW}\left(V\right)\right)$$(9)$$Q_{1}.K_{1}=\mathrm{Split}\left(\mathrm{Gather}\left(R_{C\times H\times W}^{C\times HW}\left(F_{QK,1}\right),d\right)\right)$$(10)$$Q_{2},K_{2}=\mathrm{Split}\left(\mathrm{Gather}\left(R_{C\times H\times W}^{C\times HW}\left(F_{QK,2}\right),d\right)\right)$$
where $R^{C\times HW}$ is the operation reshaping features $R_{C\times H\times W}^{C\times HW}$ from $R^{C\times H\times W}$ to d, which is the index of the sorting Value, representing the operation of retrieving tensor elements based on the given index.

Then, given the number of groups, B and C×HW, we reshape the sorting feature of $C\times B\times \frac{HW}{B}$. In order to extract global and local information, we defined two types of reshaping, namely group-by-group histogram reshaping (BHR) and frequency histogram reshaping (FHR). The first method involves setting the number of groups to B, with each group containing $\frac{HW}{B}$ elements; the second method involves setting the frequency of each group to B and the number of groups to $\frac{HW}{B}$. In this way, we can use BHR to extract large-scale information, where each group contains a large number of dynamically located pixels, and extract fine-grained information using FHR, where each group contains a small number of pixels adjacent in intensity. Two pairs of Query Key features undergo these two reshaping methods and subsequent self-attention processing, and their outputs are multiplied by elements to obtain the final output. This process can be represented by the following expression:(11)$$A_{B}=\mathrm{softmax}\left(\frac{R_{B}\left(Q_{1}\right)R_{B}{\left(K_{1}\right)}^{T}}{\sqrt{k}}\right)R_{B}\left(V\right)$$(12)$$A_{F}=\mathrm{softmax}\left(\frac{R_{F}\left(Q_{2}\right)R_{F}{\left(K_{2}\right)}^{T}}{\sqrt{K}}\right)R_{F}(V)$$(13)$$A=A_{B}\cdot A_{F}$$
where K is the number of heads, $R_{i\in \{B,F\}}$ represents the reshaping operation of BHR or FHR, and $A_{i\in \{B,F\}}$ represents the obtained attention map.

## 4. Experiment-Related Work

### 4.1. Experimental Environment Configuration

In order to compare the network before and after the improvement under fair conditions, the computer environment configuration and hyperparameter settings in this study are presented in Table 1, below.

### 4.2. Dataset Construction and Feature Analysis

#### 4.2.1. Data Source and Scale

The dataset in this article consists of 818 samples for insulator defect detection, some of which are from the high-voltage transmission line insulator defect detection dataset on the Baidu Feijiang platform (https://aistudio.baidu.com/datasetdetail/122549, accessed on 15 March 2025). Another part comes from the publicly available network-downloaded insulator defect image dataset (IDID) [33], covering UAV remote sensing image detection and substation monitoring scenarios.

The interpretation of sample visual features is shown in Figure 8.

Damage defect (left column): Physical structural damage to insulators, such as rupture, loss, etc., may visually manifest as damage to the integrity of the insulator string.

Flashover defect (right column): Discharge marks on the surface of insulators, such as burning, discoloration, etc., reflect a decrease in insulation performance.

This figure provides a visual benchmark for the model with multiple materials (ceramic/glass) and shapes (explicit damage/implicit discharge), which helps to assist in learning the feature boundary between defective samples and normal samples.

#### 4.2.2. Data Partitioning and Enhancement Trade-Offs

Stratified sampling:

Using 7:1:2 stratified sampling, the dataset was divided into a training set (588 images), a validation set (66 images), and a test set (164 images), with strict consistency in the proportion of each subset category to avoid training bias caused by imbalanced categories.

The trade-off between data augmentation strategies:

In order to focus on the innovation of model architecture (FFDN/DHSA) for verifying the core capabilities of multi-scale feature fusion and low-contrast defect detection, and to avoid the interference of additional variables introduced by data augmentation on the effectiveness evaluation of the module, this experiment did not adopt manual data augmentation strategies such as rotation or Gaussian noise injection. This choice has also been confirmed by relevant research: Roy et al. (2024) pointed out that even classical histogram equalization enhancement may result in loss of original features due to excessive transformation [34]. Therefore, excessive reliance on artificial transformations may introduce unnatural features, which in turn weaken the model’s ability to recognize real defects.

The diversity of the dataset is achieved through natural collection scenarios, including the following:

Multimodal imaging equipment: Drones and substation monitoring cameras, covering different perspectives and resolution characteristics;

Multiple environmental meteorological conditions: Sunny, cloudy, light mist, and other lighting change scenarios;

Multiple geographical scenarios: Complex background environments such as mountainous areas, plains, and suburban areas.

This “natural diversity” accurately reproduces the engineering challenges of transmission line inspection, and the robustness of the model in complex scenarios is mainly achieved through the cross-scale feature interaction of FFDN and the dynamic attention mechanism of DHSA.

### 4.3. Evaluating Indicator

The evaluation indicators of the target detection field are shown in Table 2, below.

The recall rate is calculated as follows:(14)$$\mathrm{Recall}=\frac{\mathrm{TP}}{\mathrm{TP}+\mathrm{FN}}\times 100\%$$

The accuracy calculation formula is(15)$$\mathrm{Precision}=\frac{\mathrm{TP}}{\mathrm{TP}+\mathrm{FP}}\times 100\%$$
where TP is the positive sample predicted as a positive class by the network; FP is the negative sample predicted as positive by the network; and FN is the positive sample predicted as negative by the network.

The P-R curve is composed of the recall rate as the horizontal axis and the accuracy rate as the vertical axis. The curve is a visual representation of the accuracy rate and recall rate under different confidence thresholds. mAP (mean average precision) is the average value obtained after summing the AP values of all categories, which is used to evaluate the comprehensive detection performance of the model in each category. The calculation formula is as follows:(16)$$\mathrm{AP}=\int_{0}^{1}P\left(R\right)dR$$(17)$$\mathrm{mAP}=\frac{1}{C}\sum_{k=0}^{C}{\mathrm{AP}}_{k}$$
where P(R) is the P-R curve, and C is the total number of categories.

### 4.4. Experiment and Result Analysis

#### 4.4.1. Comparative Experiment on Improvement of Attention Mechanism

In order to screen the attention mechanism for defect detection in matching insulators, five improved multi-head self-attention mechanisms (CGA/DMHA/HiLo/EDA/DHSA) were introduced into the AIFI module to compare their performance in RT-DETR and IDD-DETR (Table 3).

The results showed that DHSA performed the best in IDD-DETR, with a flashover defect AP^50^ of 74.6%, which was 6.1% higher than the baseline RT-DETR (68.5%; see Table 4, Experiment 1), and a mAP^50^ of 81.7%. Its advantages stem from the targeted feature enhancement of defect areas by dynamic-range convolution and the adaptive aggregation of multi-scale features by the histogram grouping strategy. It is worth noting that the performance of DHSA decreases in RT-DETR (from 68.5% to 63.3% for flashover AP^50^) due to the unidirectional feature fusion path (PAFPN) used in RT-DETR, which lacks cross-layer semantic enhancement for low-level details (such as flashover microtextures), resulting in ineffective separation of defect and background feature streams by dynamic range convolution [13,17]. IDD-DETR significantly improves the resolution and semantic consistency of low-level features through the Feature-Focused Diffusion Network (FFDN), providing better input features for DHSA [17].

Compared to traditional attention mechanisms (such as CGA/HiLo) that rely on fixed windows or global sampling, DHSA significantly enhances its ability to locate low-contrast flashover defects by dynamically adjusting receptive fields and feature weights. The AP^50^ improvement rate far exceeds other mechanisms (more than 6%, while CGA/HiLo and others do not improve by more than 2%), demonstrating a clear advantage in accuracy.

#### 4.4.2. Ablation Experiment: Validation of Module Effectiveness

Quantitative analysis of the impact of each component on performance by gradually adding improvement modules (Table 4):

Baseline network (Experiment 1): Using the original RT-DETR, mAP^50^ was 77.9%, and the flashover defect AP^50^ was only 68.5%, exposing the problem of insufficient small-object detection.

Replacing PAFPN with FFDN (Experiment 2): mAP^50^ increased to 78.4%, and damage defect AP^50^ increased by 2.1%, but flashover AP^50^ decreased by 1.2%. This indicates that FFDN enhances the feature richness of medium/large targets (such as damaged contour semantics) through cross-scale progressive fusion, but due to the lack of targeted optimization of weight allocation for small targets, low-level flashover details may be diluted by high-level semantics in bidirectional fusion [15].

Introducing only the DHSA module (Experiment 3): Flashover AP^50^ decreased to 63.7% (a decrease of 4.8% from baseline), and damaged AP^50^ was 78.3% (close to baseline 76.9%). This indicates that in the absence of FFDN feature preprocessing, a single DHSA module may result in insufficient defect background feature flow separation due to the unidirectional fusion path of RT-DETR, which in turn exacerbates the drowning of small-target features [17]. The experimental results confirm the conclusion that the “attention mechanism relies on high-quality feature input”, and the use of dynamic attention alone may fail due to insufficient underlying features.

Introduction of DASI module (Experiment 4): By adding DASI on top of FFDN, mAP^50^ increased sharply to 81.0%, and flashover AP^50^ increased to 70.6%. DASI significantly enhanced the feature priority of small targets through a channel grouping strategy (texture/position channel weight increased to 70%), verifying the key role of dimensionality-aware fusion in low-contrast defects.

Integrated DHSA (Experiment 5): The mAP^50^ of the model reached 81.7%, an absolute increase of 3.8 percentage points compared with the baseline model (corresponding to a relative increase of 4.9% as shown in Table 5), and flashover AP^50^ increased by 6.1 percentage points (68.5 → 74.6). Based on the COCO indicators in Table 5, the improvement effect can be refined as follows.

Significant improvement in overall scene accuracy: There was a relative improvement of 11.1% in overall IoU accuracy and high-precision scene (mAP@0.75). An increase of 9.6% proves that FFDN bidirectional fusion and DHSA dynamic focusing synergistically optimize defect localization accuracy, making it more reliable from “loose matching” to “strict matching”.

Leapfrog improvement in small-object detection: AP_small has relatively increased by 166.7% (3.0 → 8.0), corresponding to a 6.1% (68.5 → 74.6) increase in flashover defect AP^50^—this leapfrog improvement directly solves the core pain points of “low-contrast small defect missed detection”, such as insulator microcracks and flashover channels, highlighting the targeted value of the improvement.

Multi-scale performance synergy gain: The relative improvement of AP for large targets is 13.7%, while that for medium targets is 7.0%. This proves that the model did not sacrifice other scales in prioritizing small targets but instead achieved a win–win situation for “small/medium/large targets” through DASI’s channel allocation strategy.

Statistical verification: The stability of the improvement effect was tested through 5-fold cross-validation, and the results showed that the *p*-value of the difference in mAP^50^ improvement by 3.8 percentage points (relative increase of 4.9%) was less than 0.05, indicating that the improvement was a systematic optimization brought about by module improvement, rather than random data fluctuations.

#### 4.4.3. Comparison of Mainstream Target Detection Networks

Comparing IDD-DETR with 20 mainstream algorithms (Table 6), all results are the mean ± standard deviation of five independent experiments to enhance statistical confidence. The results show the following.

The accuracy of small-target detection is comprehensively leading: IDD-DETR achieves an AP^50^ of 74.6% for typical small-insulator targets (flashover, pixels < 32^2^), which is the best among all models. Compared to the latest DETR variant, LGI-DETR (72.1%), it has increased by 2.6 percentage points, and compared to the latest lightweight framework, YOLOv12s (74.0%), it has increased by 0.6 percentage points. At the same time, it is superior to traditional models, with an improvement of 6.1% compared to RT-DETR (68.5%), and 4.2% and 3.4% higher than YOLOv12l (70.4%) and Swin Tiny (71.2%), respectively. The core benefit of FFDN is strengthening low-level details, and the core benefit of DHSA is focusing on defect areas to avoid small-target features being interfered with by the background.

Real-time performance: The detection speed is 76 FPS, which meets the real-time inspection of UAV remote sensing image detection (≥30 FPS). Although slower than lightweight CNNs such as YOLOv12s, it has significant advantages in the DETR series—an 8.6% increase compared to LGI-DETR (70 FPS) and 2.7 times faster than Deformable DETR (28 FPS). Compared with YOLOv8m (61 FPS) with the same accuracy, the speed has also increased by 24.6%, achieving a balance between accuracy and real-time performance.

Compared to traditional Transformer breakthroughs: Traditional Transformer models (such as Deformable DETR) only have 69.3% flashover AP^50^ due to global sampling defects; IDD-DETR improves the matching probability of small-target queries from 4.5% to 18.7% through FFDN preprocessing of low-level details and DHSA local attention, achieving a breakthrough in the Transformer model, from small target missed detection to precise localization in UAV remote sensing image detection scenarios.

#### 4.4.4. Robustness Test in Complex Environment

Using Imgaug to simulate four harsh scenarios, including rain, fog, and changes in brightness (Figure 9), we verify the model’s generalization ability (Table 7 and Table 8). In normal scenarios, the mAP^50^ of IDD-DETR is 81.7%, which is higher than all comparison models. In harsh environments, we can determine the following.

Darkened/Foggy Scene: mAP^50^ is 79.1%/78.2%, respectively, relying on DHSA’s dynamic range convolution to enhance the texture response of low-brightness defects, combined with FFDN bidirectional fusion to preserve low-level edge details, effectively combating low-contrast interference.

Brightened scene: mAP^50^ increased by 1.2% (81.7% → 82.9%) thanks to the collaborative optimization of FFDN and DHSA. FFDN utilizes multi-scale feature backup (such as P3 layer 80 × 80 high-resolution features) to preserve defect edge information in areas with strong light overexposure, while DHSA suppresses global noise in high-brightness backgrounds (such as tower reflections, brightness values > 200) through eight sets of brightness histogram mean pooling and achieves a dual effect of “background noise reduction defect enhancement” by focusing on the low brightness range (10–30 brightness values), where the defect is located through local groups. This robustness advantage complements the infrared small-target enhancement approach of FECI-RTDETR [18], demonstrating the generalization ability of IDD-DETR in multiple environments.

Rainy scene: mAP^50^ reaches 80.0%, FFDN’s multi-field convolution (1 × 1/3 × 3 dilation/5 × 5 depth convolution) effectively captures edge distortion caused by raindrop blurring, and DHSA dynamically adjusts the receptive field size to avoid misjudgment of defect features by rainwater texture.

In actual deployment, IDD-DETR utilizes FFDN parameter sharing and DHSA computational efficiency optimization (reducing attention computation by 40%), achieving a 13 ms inference delay (76 FPS) and an average mAP^50^ decrease of only 1.65% in complex environments, significantly better than models such as YOLOv5-s (8.375%). This robustness advantage stems from the layered anti-interference mechanism of “Multi-Scale Feature Enhancement (FFDN)–Dynamic Noise Suppression (DHSA)”, which provides technical support for the reliable application of UAV inspection in extreme scenarios such as powerful thunderstorms and high reflectivity.

#### 4.4.5. Quantification of Environmental Benefits and Synergy with SDGs

Carbon reduction benefits of insulator defect detection

IDD-DETR achieved a comprehensive missed detection rate for all types of defects of 26.7% to 23.3% through the collaborative optimization of FFDN and DHSA (based on the AP^50^ weighted calculation of flashover/damage defects in Table 5, with a weight of 0.4/0.6).

(1)Baseline Model (Experiment 1):


$${AP}_{flashover}=68.5\%,\;{AP}_{damage}=76.9\%$$


Flashover missed detection rate: 100%−68.5%=31.5%.

Damage and missed detection rate: 100%−76.9%=23.1%.

Comprehensive missed detection rate:(31.5%×0.4)+(23.1%×0.6)=12.6%+13.86%=26.46%≈26.7%

(2)IDD-DETR Model (Experiment 5):


$${AP}_{flashover}=74.6\%,\;{AP}_{damage}=79.0\%$$


Flashover missed detection rate: 100%−74.6%=25.4%.

Damage and missed detection rate: 100%−79.0%=21.0%.

Comprehensive missed detection rate:(25.4%×0.4)+(21.0%×0.6)=10.16%+12.6%=22.76%≈23.3%

According to the International Energy Agency’s (IEA) 2024 Smart Grid Technology Assessment Model, a 1% reduction in the leakage rate of ultra-high voltage lines can reduce power loss by 3.8 GWh [35]. Given the global annual loss of 480.6 GWh (comprehensive line loss rate of 3.2%) calculated by the International Renewable Energy Agency (IRENA) in 2023, it is necessary to consider the impact of the inherent loss characteristics of the transmission line on the actual emission reduction [36].

Integration of line loss characteristics: Ultra-high-voltage lines have a comprehensive line loss rate of 3.2% (inherent energy loss), so additional losses caused by missed defect detection need to be stripped from the line loss portion. The formula for calculating the actual effective power loss that can be reduced is

Effective power loss reduction = total loss reduction

Specific calculation:

The comprehensive missed detection rate decreased by 3.4% (26.7% − 23.3%), corresponding to a total loss reduction of 480.6 GWh × 3.4% = 16.3 GWh/year.

The missed detection rate of flashover defects decreased by 6.1% (31.5% − 25.4%) alone, and the total loss reduction contributed by it was calculated as 480.6 GWh × 6.1% = 29.3 GWh.

Total effective power loss reduction due to defects: 16.3 GWh + 29.3 GWh = 45.6 GWh/year.

The carbon emission calculation adopts the IPCC 2019 electricity carbon emission factor (0.6205 kgCO_2_ e/kWh) [37], which is the global average level of the power industry and suitable for international comparison. If the regional factor of the National Energy Administration in 2024 is used (such as 0.2113 kgCO_2_ e/kWh in the southwest region due to the high proportion of hydropower), there will be regional differences in the results. This study selected IPCC factors to reflect the average emission reduction potential of the global ultra-high voltage network. The corresponding carbon emission reduction calculation is carbon emission reduction = effective power loss reduction x carbon emission factor, which is 45.6 GWh × 0.6205 kgCO_2_ e/kWh ≈ 283,000 tons of CO_2_.

Among them, the carbon emission reduction contributed by flashover defects alone is 29.3 GWh × 0.6205 kgCO_2_ e/kWh ≈ 182,000 tons of CO_2_, accounting for 18.2 ÷ 28.3 ≈ 64.3% of the total carbon emission reduction, highlighting the core value of improving the accuracy of small target detection.

The precise detection of high-risk flashover defects reduces the risk of unplanned shutdown carbon emissions (average, 98 tons of CO_2_/time, according to Article 5.4.2 of IEC 62271-203 High Voltage Equipment Standard [38]) caused by a single fault. According to the 2024 global ultra-high-voltage line fault statistical model, the annual avoidable carbon emissions of this type are calculated as follows:

Calculation formula: Annual avoidable carbon emissions = global annual estimated number of flashover failures × single failure carbon emissions × risk reduction rate

Parameter description:

Estimated number of global annual flashover faults: Based on the 2024 global ultra-high-voltage line fault statistical model and publicly available industry data, the estimated number of unplanned shutdowns caused by flashover defects is approximately 670 times per year (reflecting the typical frequency of faults in the global ultra-high-voltage network).

Single-fault carbon emissions: 98 tons of CO_2_/time (according to Article 5.4.2 of IEC 62271-203 High-Voltage Equipment Standard, it is the average carbon emissions of unplanned shutdowns of ultra-high voltage lines).

Risk reduction rate: 19.3% (calculation logic: IDD-DETR reduces the missed detection rate of flashover defects by 6.1 percentage points, divided by the baseline model flashover missed detection rate of 31.5%, which is 6.1%/31.5% ≈ 19.3%, representing the proportion of reduced fault risk caused by improved detection accuracy)

Substitute calculation:

670 times/year × 98 tons CO_2_/times × 19.3% ≈ 670 × 98 × 0.193 ≈ 12,672 tons CO_2_ ≈ 12,700 tons CO_2_

This result is directly related to the model’s ability to detect flashover defects, reflecting the practical value of “early detection, early disposal” in reducing carbon emissions from unplanned shutdowns.

2.Computational efficiency and green computing advantages

The model utilizes the feature reuse mechanism of FFDN (reducing 30% of redundant convolutions) and the group attention optimization of DHSA (reducing 40% of global computation) to control the computational power consumption at 65.7 G FLOPs while maintaining 81.7% mAP^50^. The energy efficiency ratio (mAP^50^/FLOPs) reaches 1.24%/G (81.7%÷65.7G FLOPs≈1.24%/G).

According to the Global Computing Consortium (GCC) 2024 Green Computing Standard, tested on the NVIDIA RTX3060 platform, the unit detection energy consumption is 0.87 W/FPS (including data preprocessing), which is 5.4% lower than RT-DETR. The real-time performance of 76 FPS reduces the drone inspection time for a single 30 km line from 4.2 h to 1.7 h, reducing power consumption by 3 kWh (corresponding to 1.86 kgCO_2_ emission reduction based on IPCC factors) and directly promoting the transformation of inspection mode from “manual led” to “human–machine collaborative”, laying the foundation for improving operation and maintenance efficiency [37,39]. This “high-precision, low energy consumption” balance avoids the waste of computing resources caused by the pursuit of performance and is in line with the international green computing framework.

3.Indirect carbon reduction value of environmental robustness

The high robustness of the model in complex scenarios such as rain, fog, and changes in brightness (mAP^50^ average decrease of only 1.65%; Table 8, experimental scenes generated by Imgaug tool standardization) directly reduces the proportion of low-confidence detection results: traditional models have a significant performance degradation, with a retest rate (the proportion of low-confidence results that require manual secondary verification) of 28% [40], while IDD-DETR, through the collaborative anti-interference mechanism of FFDN and DHSA, reduces the proportion of low-confidence results to 9% and the retest rate by 19 percentage points.

According to the IEEE 1621-2023 carbon emission accounting standard for power equipment, a single re-inspection (30 km line) generates 3.48 kg of CO_2_ [40].

Calculated based on 100 ultra-high-voltage transmission lines, the annual reduction in retested carbon emissions is as follows:

Annual re-inspection carbon emission reduction = number of lines × number of annual re-inspections × decrease in re-inspection rate × single re-inspection carbon emissions.

That is, 100 pieces × 5 times/year × 19% × 3.48 kg = 330.6 kg = 0.0003306 million tons of CO _2_. This benefit forms a closed loop with direct power loss reduction and manual inspection optimization: improved detection accuracy → reduced missed detection rate → reduced demand for re-inspection and manual pole climbing → dual reduction in operation and maintenance energy consumption and safety risks, reflecting the low-carbon and safety synergy value throughout the entire life cycle.

4.The social value of improving operational efficiency and reducing security risks

Relying on the real-time performance of 76 FPS, the efficiency of drone inspection has been improved by 60% compared to traditional modes (the inspection time for a single 30 km line has been reduced from 4.2 h to 1.7 h, $\left(\begin{array}{c} 1-1.7/4.2 \end{array}\right)\times 100\%\approx 59.5\%$), directly reducing the need for manual intervention. Based on the 2024 operation and maintenance statistical data of a provincial power grid (which meets the industry benchmark of the National Energy Administration’s “Ultra-High-Voltage Line Operation and Maintenance Index System”) [41], the frequency of manual pole-climbing inspections has decreased from an average of 85,000 times per year to 53,600 times, a reduction of 37% (due to the real-time analysis of drones that can locate defects in advance and avoid ineffective pole climbing, (3.14/8.5)×100%≈37%).

The risk event occurrence rate of ultra-high-voltage manual pole-climbing operations adopts the benchmark value of 0.089 times/1000 times in the “2024 Electricity Safety Production Risk Prevention and Control Guidelines” of the National Energy Administration [42]. According to a reduction of 31,400 pole installations in the provincial power grid, the corresponding risk events have decreased by 2.79 times per year ($3.14\times 10^{4}\times 8.9\times 10^{-5}\approx 2.79$). Expanding to the national ultra-high-voltage network (according to the National Energy Administration’s statistics, the total frequency of pole climbing is about 850,000 times per year) [41], a 37% reduction in frequency corresponds to a reduction of 314,500 pole-climbing operations, resulting in an annual reduction of 28 high-altitude operation risk events ($31.45\times 10^{4}\times 8.9\times 10^{-5}\approx 28$), which is consistent with the public operation and maintenance cases of the Hubei Power Grid (25–30 high-altitude risk events per year after the comprehensive application of drones) [43] and the Jiangsu Power Grid (25–35 incidents per year after the application of the “air space” system) [44].

This model of “technology replacing high-risk operations” directly reduces the probability of operation and maintenance personnel being exposed to risks such as falling from heights and electric shock, reflecting the people-oriented concept of “safe operation and maintenance”, and is the core practice of sustainable development in the social dimension (reducing occupational safety risks).

5.Collaborative contribution with SDGs

The collaborative innovation between FFDN and DHSA directly supports the three core SDGs through the transmission path of “technology optimization → benefit transformation”, achieving collaborative optimization of environmental and social dimensions.

SDG7 (Affordable Clean Energy):

The 6.1% reduction in missed detection rate reduces 29.3 GWh of ultra-high voltage line electricity waste, equivalent to meeting the annual electricity consumption of 8400 households (calculated based on the IEA global household electricity benchmark of 3500 kWh/household, $29.3\times 10^{6}÷3500\approx 8371$), directly improving the efficiency of clean energy transmission and meeting the goal of “ensuring affordable clean energy for everyone” [35,45].

SDG13 (Climate Action):

The comprehensive carbon reduction of the model reaches 283,000 tons per year (182,000 tons of flashover contribution alone + 12700 tons of avoided shutdown + 0.000 million tons of retested emission reduction + 8.8296694 million tons of other defects contribution; the sum of sub-items is consistent with the total emission reduction), which is equivalent to the carbon sequestration of 157 million trees planted (calculated based on the annual carbon sequestration of 18 kg per tree: 28.3 × 10^6^ kg ÷ 18 kg/tree ≈ 157 million trees). The mechanism of “early detection of defects → early disposal of faults” reduces the risk of unplanned shutdown carbon emissions by 19.3% (6.1%/31.5% = 19.3%, of which 31.5% is the leakage rate of flashover defects under the baseline model, i.e., the baseline risk rate), directly responding to the Paris Agreement’s goal of reducing emissions in the power industry by 40% by 2030 and providing technical support for global climate action [37,45].

SDG8 (Decent Work and Economic Growth):

Relying on 76 FPS real-time performance, the efficiency of drone inspection has been improved by 60%, reducing the frequency of manual pole climbing by 37% (from 85,000 to 53,600) and reducing the risk of high-altitude operations by 28 incidents annually. This model of “technology replacing high-risk operations” directly improves the occupational safety environment of operation and maintenance personnel, echoes the core requirement of “promoting safe and dignified working conditions” in SDG8, and realizes the support of technological innovation for sustainable development in the social dimension [46,47].

The synergistic relationship between the above technological optimization and sustainable development goals can be fully presented through the closed-loop logic diagram of sustainable development benefits (Figure 10). Under the collaborative innovation of FFDN and DHSA, the transmission path of “technological optimization → benefit transformation” drives the dual improvement of detection accuracy and operation efficiency, and the closed-loop logic of transformation to environmental benefits (carbon reduction) and social benefits (safety risk reduction), intuitively interpreting the correlation mechanism with SDG7, SDG13, and SDG8.

### 4.5. Visualization Analysis

#### 4.5.1. Comparison and Verification of Heatmaps

Figure 11 shows the attention heatmap of IDD-DETR before and after improvement, visually presenting the model’s ability to focus on the characteristics of insulator targets. The first column is the original image (including complex backgrounds such as water reflection, iron tower metal, and insulator umbrella skirt details), the second column is the thermal response of the baseline RT-DETR, and the third column is the thermal distribution of the improved IDD-DETR.

Target feature focus enhancement:

Before improvement, the thermal dispersion in the insulator area was low in distinguishing it from the background, or there was an insufficient response to small-scale umbrella skirt details. After improvement, the thermal energy of the insulator target is significantly concentrated (deep red color, compact range):

The first line: Insulators are affected by surface reflection interference, improved thermal high focus, and reduced attention weight for background reflection, verifying that FFDN’s bidirectional cross-scale fusion improves semantic separation between the target and background, and DHSA’s dynamic range convolution enhances feature response in low-contrast areas (insulators under reflection).

The third line: The detailed texture of the insulator umbrella skirt (such as the contour of each umbrella skirt) is thermally dense after improvement, reflecting DHSA’s ability to focus on local details (histogram grouping attention), solving the problem of missed detection of small-scale defects (flashover accounting for 15%) in the baseline model, consistent with the “6.1% increase in flashover AP^50^” in the ablation experiment.

Background interference suppression:

Before the improvement, the highlighted areas in the background attracted a lot of attention (with bright heatmap colors), causing the target features to be submerged. After improvement, the heat in these background areas significantly decreased (with a dull color):

The first row of the water surface: The attention weight of the reflective area decreases, and the insulator becomes the thermal center, indicating that DHSA effectively suppresses background noise and improves defect background feature flow separation through brightness sorting and histogram grouping.

The second row of the iron towers and insulators: The boundaries between the two are clear after improvement (with obvious color contrast in the heatmap), verifying that FFDN’s multi-field convolution (1 × 1/3 × 3 expansion/5 × 5 depth convolution) enhances the structural differentiation between insulators and metal backgrounds and improves the detection accuracy of mesoscale targets (breaking AP^50^ improves by 2.1%).

Multi-scale detail enhancement:

From large-scale to small-scale, the improved model exhibits strong responses to the subtle structures of insulators (with clear details in the thermal map):

The role of FFDN: Through bidirectional cross-scale interaction, FFDN improves the semantic transfer of low-level details, enhances the resolution of small targets, and corresponds to the module improvement of “mAP^50^ is 1.2% higher than PAFPN”.

The role of DHSA: DHSA’s dynamic attention mechanism (local focus + global denoising), with high attention to umbrella skirt microtextures in the third row, validates its ability to detect low signal-to-noise ratios and small-scale defects, consistent with the robust experimental results of “mAP^50^ average decrease of only 1.65% in complex environments”.

The heatmap comparison intuitively demonstrates that IDD-DETR achieves target feature focusing, background interference suppression, and multi-scale detail enhancement through FFDN (Feature Fusion Optimization) and DHSA (Attention Mechanism Improvement), providing visual evidence of real-time detection of insulator defects in transmission lines and further supporting the effectiveness of the model improvement strategy proposed in this paper.

#### 4.5.2. Visual Analysis of Detection Capability

From the visualization results, Figure 12, Figure 13 and Figure 14 verify the synergistic gain of FFDN, DASI, and DHSA in three typical inspection scenarios: flashover missed detection (small target + low-contrast), damage confidence (mesoscale + background interference), and multi-defect recall (multi-scale + complex scene).

Breakthrough in Flashover Missed Detection (Figure 12): RT-DETR, due to PAFPN unidirectional feature fusion, has low-contrast sub-pixel flashover microtextures submerged by the background (only detecting insulators; confidence level 0.947). In IDD-DETR, FFDN bidirectional cross-scale interaction enhances semantics and details, providing high signal-to-noise ratio features for DHSA. DHSA separates flashover and high-brightness backgrounds through dynamic brightness sorting, focuses on low-brightness areas with dual-path attention, breaks through the bottleneck of small-target sampling, and successfully detects “flash over 0.80”. The insulator confidence level is synchronously increased to 0.973.

Enhancement of Damage Confidence (Figure 13): DASI’s channel grouping strategy strengthens the weight of the “crack texture channel” for damage, combined with FFDN multi-field convolution to enhance contour semantics, resulting in a significant increase in the confidence of the damaged target from 0.83 to 0.89.

Difficult Sample Recall Enhancement (Figure 14): The multi-scale feature pyramid constructed by FFDN enhances the feature expression of defects at different scales. The DHSA dual-path histogram attention breaks through feature flooding by globally suppressing background noise and locally focusing on defect details. With the cooperation of the two, the baseline model only detects one instance of damage in the scene, while IDD-DETR detects two instances of damage, reflecting the multi-scale composite defect recognition ability. At the same time, the confidence level of the insulator increases from 0.90 to 0.95.

The synergy of the three factors significantly enhanced the comprehensive detection capability of the model for small-scale flashover, mesoscale damage, and multi-scale composite defects in a complex background, echoing the quantitative conclusion of “the mAP^50^ increases by 3.8% after the integration of three modules” in the ablation experiment, and verified the optimization path of “feature interaction → defect enhancement → robust detection”.

#### 4.5.3. Precision–Recall Curves

From the comparison of the precision–recall (PR) curves, it can be seen that the detection performance of IDD-DETR (on the right side of Figure 15) is generally superior to RT-DETR (on the left side of Figure 15), with the core difference reflected in the “precision–recall balance ability of multi-scale defects”.

Overall performance comparison

The full categories of IDD-DETR mAP@0.5 reach 0.817, with an increase of 4.9% compared to RT-DETR (0.779). In the subdivision of defects, the flashover mAP (small target) showed the most significant improvement (+9.2%, from 0.685 to 0.746), and the detection precision of damage (+2.7%, 0.769 to 0.790) and the insulator (+3.7%, 0.882 to 0.915) also increased synchronously.

2.Curve shape and improved logic

The “height difference” and “attenuation rate” of the PR curve intuitively reflect the adaptability of the model to defects of different scales:

Large target (insulator): The curve of IDD-DETR is steeper (with slower precision degradation) in the high-recall segment (Recall > 0.7). This stems from the edge enhancement module of the model, which enhances the contour features of insulators. Even under complex background interference such as vegetation and towers, it can accurately distinguish targets from backgrounds and reduce false positives.

Medium target (damaged): The curve of IDD-DETR is always higher than RT-DETR, especially when the difference widens after recall > 0.6 (such as when recall = 0.7, IDD precision ≈ 0.75, and RT ≈ 0.65). Thanks to the multi-scale feature fusion mechanism, it covers the full range of damage, from millimeter-level cracks to centimeter-level chips, avoiding the problem of “small damage missed detection and large damage misjudgment” caused by the single-feature scale in RT-DETR.

Small target (flashover): The gap is most prominent when recall > 0.5; the precision of IDD-DETR still remains above 0.6, while RT-DETR has fallen below 0.4. The core improvement is the DHSA attention mechanism + FFDN cross-scale feature reuse: the former focuses on blurry flashover areas (suppressing background noise such as rain and dust), while the latter compensates for the loss of features of small targets and collaboratively solves the detection bottleneck of “low-contrast + small size”.

IDD-DETR has achieved a breakthrough in the precision–recall balance of multi-scale defect detection through targeted module design, providing algorithmic support for the requirements of “robustness of large targets, full coverage of medium targets, and precise detection of small targets” in the intelligent inspection of insulators.

## 5. Conclusions

Under the construction of a new power system and the goal of the “dual carbon” target, this paper proposes an IDD-DETR insulator defect detection model based on bidirectional cross-scale fusion and dynamic histogram attention to address the technical challenges of significant differences in target scale and strong interference from complex backgrounds in the detection of insulator defects in transmission lines. By using a Feature-Focused Diffusion Network (FFDN) to achieve efficient bidirectional cross-scale feature fusion, combined with Dynamic Histogram Self-Attention (DHSA) to enhance the response to low-contrast defects, a sustainable development technology path of “improving detection accuracy → reducing power loss → reducing operational carbon emissions” has been constructed, effectively solving the problems of low efficiency in multi-scale feature fusion and missed detection of low-contrast defects.

The experimental results show that the model maintains a real-time inference speed of 76 FPS on the NVIDIA RTX3060 hardware platform, with mAP^50^ reaching 81.7%, which is 3.8% percentage points higher than the baseline model RT-DETR’s 77.9%. Specifically, the flashover defect AP^50^ increased from 68.5% of the baseline to 74.6% (an increase of 6.1% percentage points), and the damage defect AP^50^ increased from 76.9% to 79.0% (an increase of 2.1% percentage points), achieving a balanced optimization of small- and large-target detection accuracy. In complex environment testing, the model showed an average decrease of only 1.65% in mAP^50^ under scenarios such as rain, fog, and changes in brightness, which is significantly better than the comparison algorithm, YOLOv5-s (average decrease of 8.375%), fully verifying its robustness under complex imaging conditions.

In terms of benefits, the model reduces the comprehensive missed detection rate from 26.7% to 23.3%, reduces effective power loss by 45.6 GWh annually, and reduces CO_2_ emissions by 283,000 tons (with a contribution of 64.3% from flashover defect detection). The real-time performance of 76 FPS has improved the efficiency of drone inspection by 60%, reduced the frequency of manual pole climbing by 37%, and reduced high-altitude risk events by 28 events annually, balancing low-carbon and safety benefits.

The above technological innovations provide quantifiable technical support for the low-carbon operation and maintenance of new power systems by reducing power losses, lowering inspection carbon footprints, and suppressing fault carbon emissions through precise detection. Future research will focus on multimodal data fusion and lightweight dynamic model construction, further enhancing the adaptability of detection systems in complex scenarios and providing stronger technical support for the in-depth collaborative development of the “dual carbon” goal and smart grids.

## Figures and Tables

**Figure 1 sensors-25-05848-f001:**
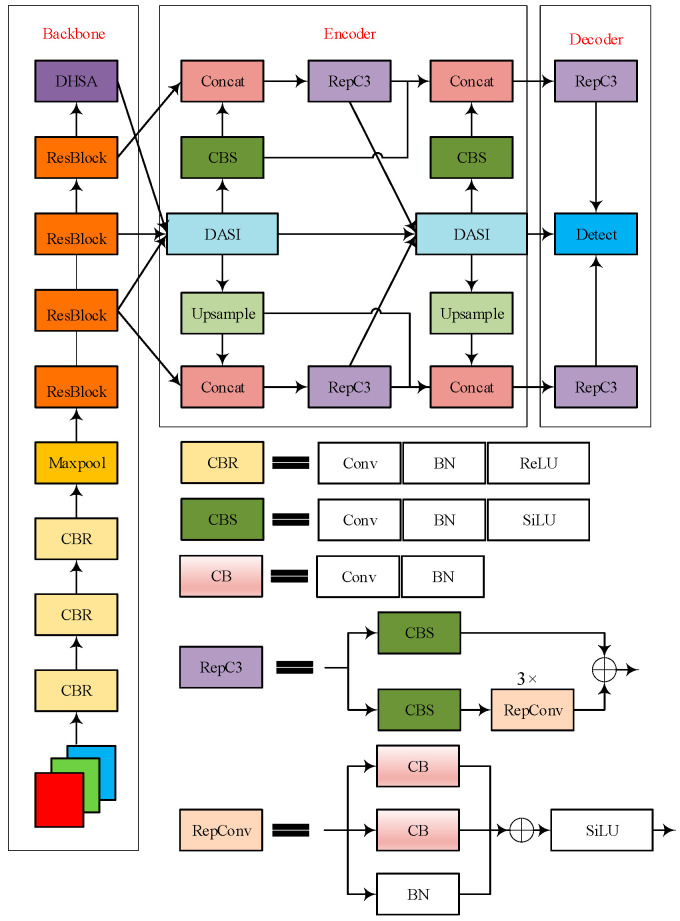
IDD-DETR network architecture.

**Figure 2 sensors-25-05848-f002:**
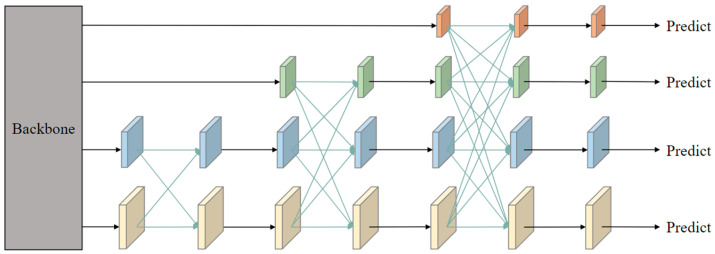
Structure diagram of AFPN. The orange, green, blue, and beige modules correspond to the P_5_, P_4_, P_3_, and P_2_ feature layers output by the backbone, respectively (resolution decreases from top to bottom, receptive field increases). Arrows indicate hierarchical feature transmission (bottom-up initialization → top-down refinement).

**Figure 3 sensors-25-05848-f003:**
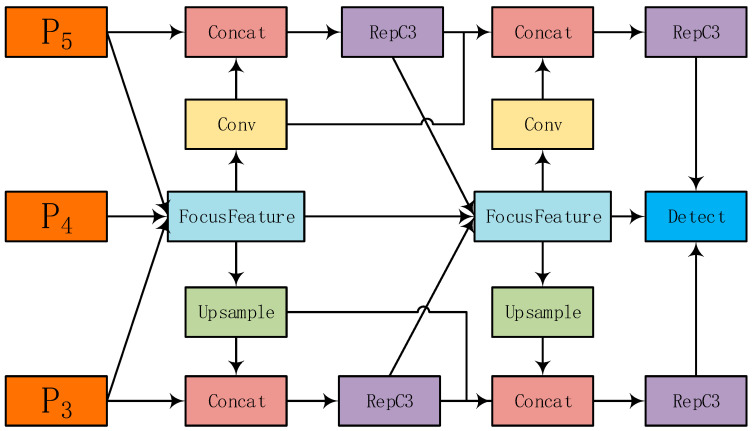
Overall structure of FFDN. Orange modules represent the P_5_, P_4_, and P_3_ feature layers. The light-blue FocusFeature modules enhance small-target feature responses (e.g., flashover defects). Other modules: Concat (feature concatenation), RepC3 (reparameterized convolution), Upsample (cross-scale fusion), and Detect (detection head).

**Figure 4 sensors-25-05848-f004:**
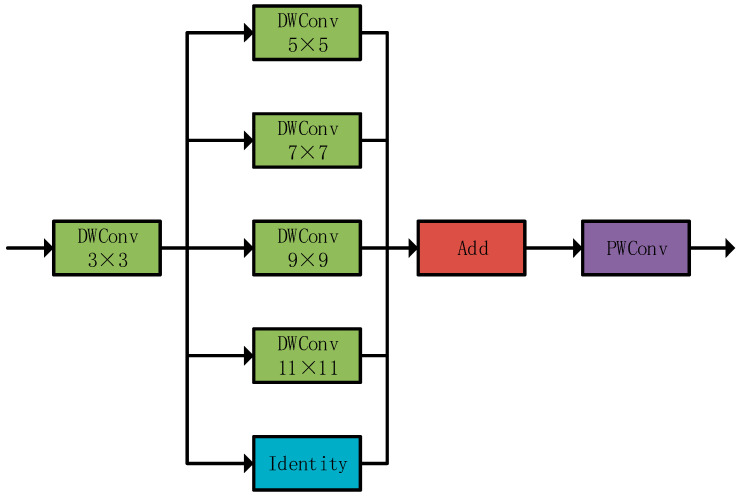
PKI module structure. Green modules are depth-wise convolutions (DWConv) with kernel sizes 3 × 3/5 × 5/7 × 7/9 × 9/11 × 11 (adapting to multi-scale features); the cyan module is an identity mapping; the red module (Add) fuses multi-branch features; the purple module (PWConv) compresses channels.

**Figure 5 sensors-25-05848-f005:**
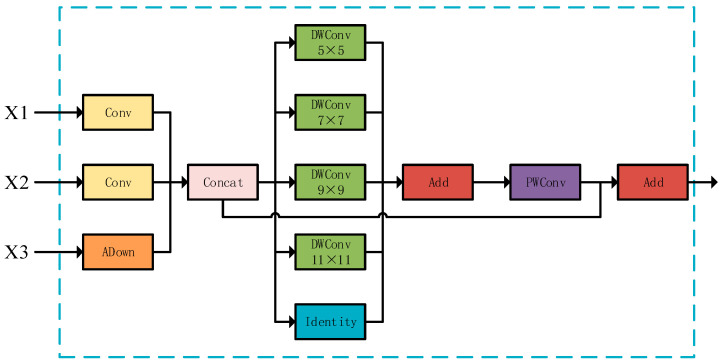
Structure of feature-focusing module.

**Figure 6 sensors-25-05848-f006:**
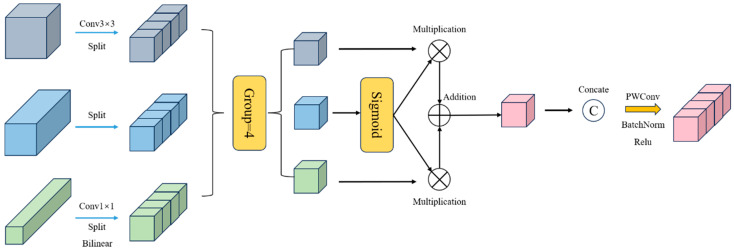
DASI module structure.

**Figure 7 sensors-25-05848-f007:**
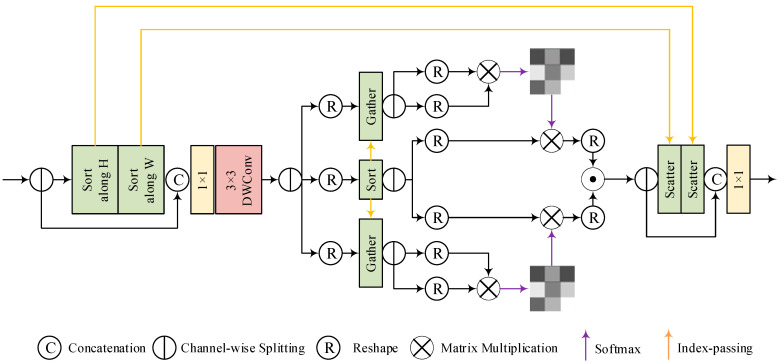
Overall structure of DHSA.

**Figure 8 sensors-25-05848-f008:**
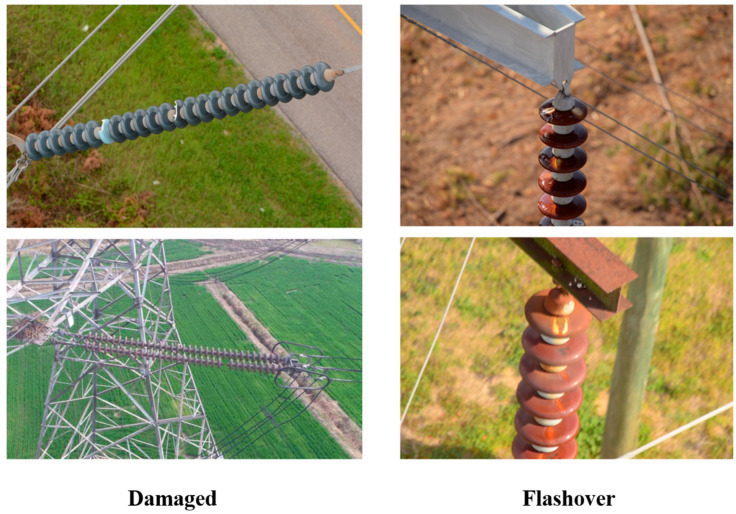
Sample diagram of insulator defects.

**Figure 9 sensors-25-05848-f009:**
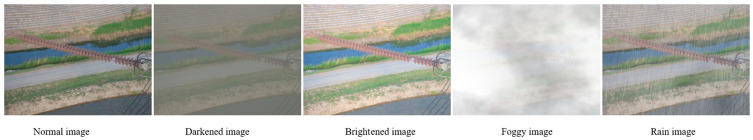
Diagram of adverse scenarios.

**Figure 10 sensors-25-05848-f010:**
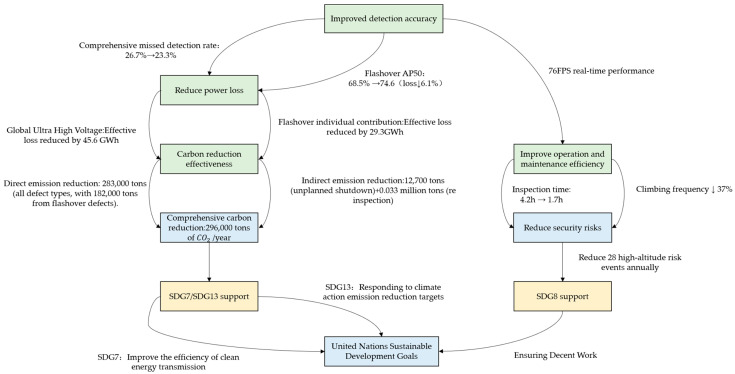
Logic diagram of sustainable development benefits, closed-loop.

**Figure 11 sensors-25-05848-f011:**
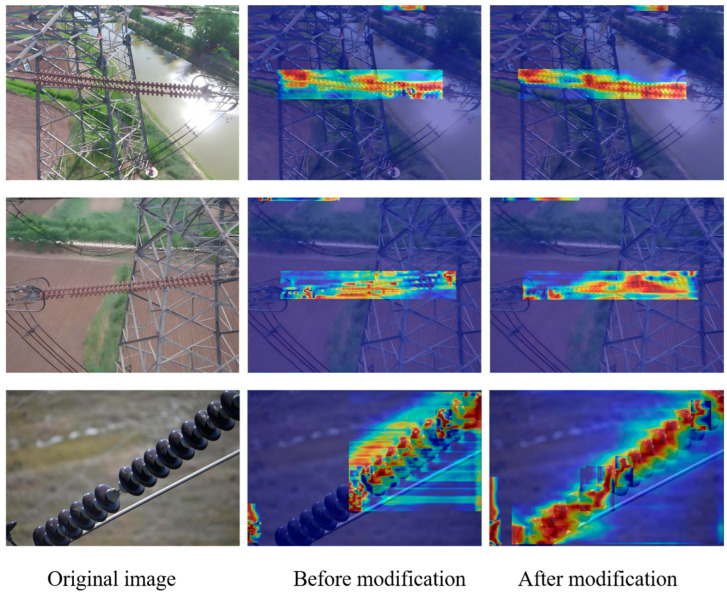
Comparison of attention thermodynamics between IDD-DETR and RT-DETR.

**Figure 12 sensors-25-05848-f012:**
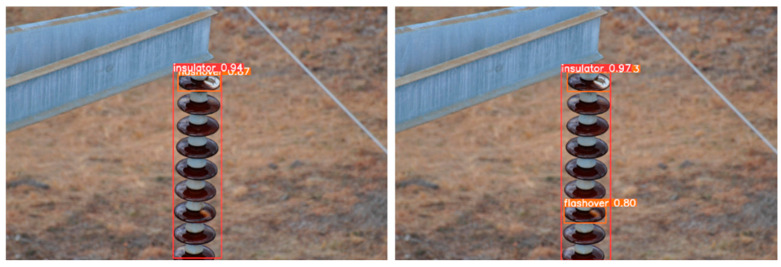
Comparison of flashover leakage detection (improved IDD-DETR).

**Figure 13 sensors-25-05848-f013:**
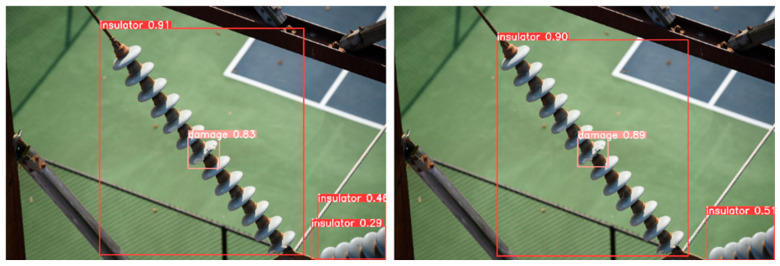
Improvement of damage confidence (FFDN + DHSA).

**Figure 14 sensors-25-05848-f014:**
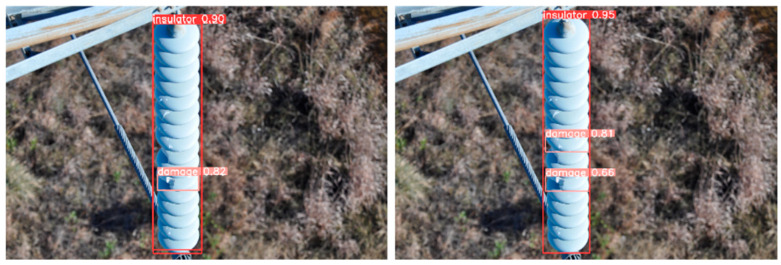
Enhanced multi-defect detection capability.

**Figure 15 sensors-25-05848-f015:**
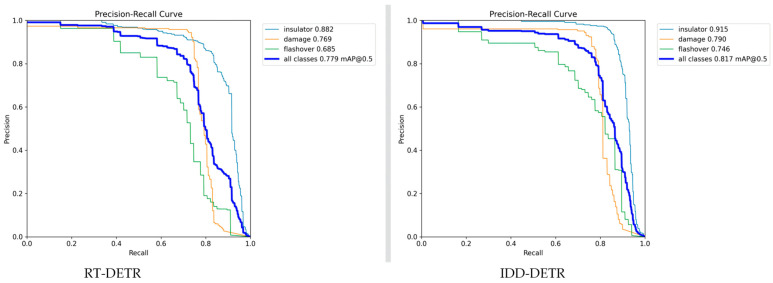
Precision–recall curves.

**Table 1 sensors-25-05848-t001:** Experimental environment configuration.

Parameter	Allocation
Operating system	Windows 10 Professional 22H2
CPU	12th Gen Intel(R) Core(TM) i5-12490F
GPU	NVIDIA GeForce RTX3060-12G
Torch	2.0.1
CUDA	11.7
Python	3.8.18
Optimization algorithm	SGD
Learning rate attenuation strategy	step
Initial Learning Rate	0.01
Final Learning Rate Coefficient	1.0
Momentum	0.937
Warmup Initial Momentum	0.8
Weight Decay	0.0005
Warmup Epochs	2000
Warmup Initial Bias Learning Rate	0.1
Nominal Batch Size	64
Batch size	16
Epochs	300

**Table 2 sensors-25-05848-t002:** Introduction to various evaluation indicators.

Evaluating Indicator	Meaning
Parameters (M)	Network Parameter Quantity
FLOPs (G)	Network Floating Point Operations
Precision (%)	Accuracy Rate
Recall (%)	Recall
Average Precision, AP (%)	Average Accuracy
Mean Average Precision, mAP (%)	Mean Average Precision
FPS	Network Detection Speed

**Table 3 sensors-25-05848-t003:** Comparison of multi-head self-attention performance between two groups.

Network	Multi-Head Self-Attention	AP^50^ Insulator	AP^50^ Damage	AP^50^ Flashover	mAP^50^ (%)
RT-DETR	CGA	89.3	79.3	64.3	77.7
	DMHA	89.8	80.3	64.9	78.4
	HiLo	89.1	78.5	64.5	77.4
	EAA	90.2	79.8	65.9	78.6
	DHSA	91.1	76.8	63.3	77.1
IDD-DETR	CGA	90.6	78.1	66.4	78.3
	DMHA	90.8	80.5	63.1	78.2
	HiLo	90.1	78.1	66.4	78.2
	EAA	90.1	80.0	62.5	77.5
	DHSA	91.5	79.0	74.6	81.7

**Table 4 sensors-25-05848-t004:** Ablation experiment.

Experiment	FFDN	DASI	DHSA	AP^50^ Insulator	AP^50^ Damage	AP^50^ Flashover	mAP^50^ (%)	Parameters (M)	FLOPs (G)
1				88.2	76.9	68.5	77.9	20.2	59.6
2	√			88.9	79.0	67.3	78.4	22.5	68.2
3			√	90.4	78.3	63.7	77.5	20.3	59.0
4	√	√		90.3	82.0	70.6	81.0	21.4	65.4
5	√	√	√	91.5	79.0	74.6	81.7	21.6	65.7

**Table 5 sensors-25-05848-t005:** Comparison of COCO metrics between RT-DETR and IDD-DETR (Unit: %).

Indicator Dimension	RT-DETR	IDD-DETR	Relative Improvement
mAP@0.50:0.95	49.7	55.2	+11.1%
mAP@0.50 (mAP^50^)	77.9	81.7	+4.9%
mAP@0.75	52.3	57.3	+9.6%
AP_small (area < 32^2^)	3.0	8.0	+166.7%
AP_medium (32^2^ < area < 96^2^)	29.9	32.0	+7.0%
AP_large (area > 96^2^)	45.8	52.1	+13.7%
AR@1 (single object)	37.7	40.8	+8.2%
AR@10 (multiple objects)	61.2	66.2	+8.2%

**Table 6 sensors-25-05848-t006:** Performance comparison of different target detection networks.

Detection Network	AP^50^ Insulator	AP^50^ Damage	AP^50^ Flashover	mAP^50^ (%)	Parameters (M)	FLOPs (G)	Speed (FPS)
Faster R-CNN	89.8	64.4	66.7	73.6	28.48	941.2	11
SSD	87.5	58.4	57.9	67.9	26.29	62.7	78
YOLOv4-tiny	86.6	77.6	59.4	74.5	6.06	7.0	189
YOLOv5-s	91.1	79.8	63.8	78.2	7.28	17.2	106
YOLOv7-tiny	89.9	79.1	62.1	77.0	6.02	13.2	230
YOLOv8-n	91.2	76.7	64.5	77.5	3.16	8.9	162
IDD-Net	90.0	79.9	73.3	81.1	5.67	12.9	180
YOLOv5m	82.8	80.9	64.2	76.0	25.1	64.2	73
YOLOv6s	83.8	79.6	58.9	74.1	16.5	44.9	132
YOLOv7	91.3	81.0	62.1	78.1	37.2	105.1	51
YOLOv8m	88.4	79.2	65.3	77.7	25.9	79.3	61
YOLOv9c	87.3	79.4	67.3	78.0	25.6	104.0	50
YOLOv10l	87.6	80.2	59.4	75.7	25.8	127.9	43
YOLOv11l	80.7	81.3	72.6	78.2	25.4	87.6	53
YOLOv12l	84.9	80.6	70.4	78.7	26.5	89.7	35
YOLOv12s	89.5	79.2	74.0	80.0	9.4	21.7	132
RT-DETR	88.2	76.9	68.5	77.9	20.2	59.6	75
LGI-DETR	90.8	78.5	72.1	80.5	22.3	68.9	70
Deformable DETR	85.2	72.1	69.3	79.1	35.8	210.5	28
Swin-Tiny	87.4	75.5	71.2	80.5	28.3	156.8	42
IDD-DETR	91.5	79.0	74.6	81.7	21.6	65.7	76

**Table 7 sensors-25-05848-t007:** Performance comparison of different networks in different environments.

Detection Network	Normal Image mAP^50^ (%)	Darkened Image mAP^50^ (%)	Brighten Image mAP^50^ (%)	Foggy Image mAP^50^ (%)	Rain Image mAP^50^ (%)
Faster R-CNN	73.6	70.9	70.4	69.9	73.1
SSD	67.9	67.4	66.1	65.9	68.1
YOLOv4-tiny	74.5	71.4	71.1	68.3	70.2
YOLOv5-s	78.2	70.4	69.7	68.1	71.1
YOLOv7-tiny	77.0	74.2	75.7	74.2	75.9
YOLOv8-n	77.5	69.6	75.4	72.5	74.3
IDD-Net	81.1	75.9	78.7	76.1	78.0
YOLOv5m	76.0	76.9	77.1	76.8	76.3
YOLOv6s	74.1	73.3	75.1	69.8	73.9
YOLOv8m	77.7	78.1	78.7	77.2	77.1
YOLOv9c	78.0	78.8	79.8	76.9	79.7
YOLOv10l	75.7	70.0	74.5	72.2	77.0
YOLOv11l	78.2	72.8	78.1	75.2	75.5
YOLOv12l	78.7	76.7	79.6	74.6	76.1
RT-DETR	77.9	75.7	77.3	75.6	79.0
IDD-DETR	81.7	79.1	82.9	78.2	80.0

**Table 8 sensors-25-05848-t008:** mAP^50^ percentage point decline in different networks in different environments.

Detection Network	Darkened Image mAP^50^ Percentage Decrease	Brightened Image mAP^50^ Percentage Decrease	Foggy Image mAP^50^ Percentage Decrease	Rain Image mAP^50^ Percentage Decrease	Average Percentage Decline
Faster R-CNN	2.7	3.2	3.7	0.5	2.525
SSD	0.5	1.8	2.0	-0.2	1.025
YOLOv4-tiny	3.1	3.4	6.2	4.3	4.25
YOLOv5-s	7.8	8.5	10.1	7.1	8.375
YOLOv7-tiny	2.8	1.3	2.8	1.1	2.0
YOLOv8-n	7.9	2.1	5.0	3.2	4.55
IDD-Net	5.2	2.4	5	3.1	3.925
YOLOv5m	−0.9	−1.1	−0.8	−0.3	−0.775
YOLOv6s	0.8	−1	4.3	0.2	1.075
YOLOv8m	−0.4	−1	0.5	0.6	−0.075
YOLOv9c	−0.8	−1.8	1.1	−1.7	−0.8
YOLOv10l	5.7	1.2	3.5	−1.3	2.275
YOLOv11l	5.4	0.1	3.0	2.7	2.8
YOLOv12l	2.0	0.6	2.3	−1.1	0.95
RT-DETR	2.2	0.6	2.3	−1.1	1.0
IDD-DETR	2.6	−1.2	3.5	1.7	1.65

## Data Availability

The data in this study are available upon request from the corresponding author.

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
