# Peer review of "IDD-DETR: Insulator Defect Detection Model and Low-Carbon Operation and Maintenance Application Based on Bidirectional Cross-Scale Fusion and Dynamic Histogram Attention"

_sensors, 2025, doi:10.3390/s25185848_

Round 1
Reviewer 1 Report (Previous Reviewer 2)
Comments and Suggestions for Authors
The author has well revised and responded to the previous comments and questions, and the quality of the paper has been improved. Now this paper can be considered for acceptance.
Author Response
Sincerely thank you for your meticulous review and valuable suggestions on the paper! Your professional opinion provides crucial guidance for in-depth research, from scenario-based argumentation in the introduction to method optimization. Every modification reflects your pursuit of academic rigor. Special thanks to everyone for their affirmation and recommendation in the final review. This is not only a recognition of our work but also a driving force for us to continuously deepen our cultivation. I would like to express my sincerest gratitude once again! Looking forward to receiving your guidance and assistance in the future!
Reviewer 2 Report (Previous Reviewer 1)
Comments and Suggestions for Authors
In this article, an improved network IDD-DETR is proposed for Insulator defect detection. Some defect image and result image of detection are supplemented in the article. The proposed solution is helpful for construction of a new power system and the drive of the "dual carbon" target. I think if the figures are improved, the article will be better, for example, fig.2-4.
Author Response
Sincerely thank you for your meticulous review and constructive suggestions! We highly appreciate your insights on enhancing figure clarity. Following your advice, we have supplemented detailed captions for Fig. 2–4, explicitly explaining module functions, feature layer meanings to improve the interpretability of the architectural diagrams. Your guidance is invaluable for refining our work. We remain committed to polishing the manuscript and welcome any further feedback.
Reviewer 3 Report (New Reviewer)
Comments and Suggestions for Authors
The paper titled “IDD-DETR: Insulator Defect Detection Model and Low Carbon 2 Operation and Maintenance Application Based on Bidirec- 3 tional Cross-Scale Fusion and Dynamic Histogram Attention” proposes IDD-DETR, an RT-DETR variant integrating a Feature Focused Diffusion Network (FFDN), Dimension-Aware Selective Integration (DASI), and Dynamic Histogram Self-Attention (DHSA) for UAV-based insulator defect detection. While the topic is relevant, the work suffers from limited novelty, insufficient experimental rigor, and weak positioning against the state-of-the-art.
Major Concerns:
- The proposed modules adapt existing concepts (e.g., AFPN, PKI, histogram-based attention) with minimal architectural innovation. The contribution reads as an incremental improvement rather than a fundamental advance in detection methodology.
- The dataset is small (818 images, with only 123 flashover defects) and lacks diversity control, raising overfitting concerns. With this scale, the problem could likely be addressed using simpler CNN-based methods, as demonstrated in:
- Automatic Generation of Laser Cutting Paths in Defective TFT-LCD Panel Images by Using Neutrosophic Canny Segmentation
- Mathematical Analysis of Histogram Equalization Techniques for Medical Image Enhancement: A Tutorial from the Perspective of Data Loss
- Many figures are blurry and hard to interpret.
- No cross-dataset evaluation or external validation is provided.
- Reported gain (+3.8% mAP50) is not supported by statistical significance testing. Only mAP50 is reported standard COCO mAP, AP_small/medium/large, and per-IoU breakdowns are missing, making it hard to assess stability, especially for small-object claims.
- No precision–recall curves are included.
- Hyperparameters for modules, histogram binning strategy, and other key parameters are not disclosed, hindering reproducibility.
- The “low-carbon” and environmental benefits are speculative and based on unverified assumptions rather than measured outcomes.
- The paper omits comparisons with several high-performing DETR variants and small-object detection frameworks.
- The manuscript is verbose, with marketing-style language in technical sections, which obscures the core contributions.
Round 2
Reviewer 3 Report (New Reviewer)
Comments and Suggestions for Authors
The authors have answered most of my concerns, but I remain unconvinced by their response to the following statement in the manuscript:
“It is worth noting that some traditional methods have shown advantages in specific scenarios, such as edge extraction technology based on Neutrosophic Canny segmentation, which can achieve precise localization of small sample defects in structured scenarios (such as TFT-LCD panel defects) through refined operator design [10]. However, in complex background (multi material, multi environmental interference) scenarios such as transmission line insulators, their adaptability is limited and they are difficult to cope with dynamic inspection environments.”
My concern is that this wording is confusing and somewhat inconsistent with the scope of the present paper. The authors note that edge-based methods struggle with insulator inspection due to complex backgrounds. Yet, the contribution of this paper is precisely in detecting and segmenting such complex insulator defects, which are arguably more challenging than TFT-LCD cases. Furthermore, prior TFT-LCD studies also emphasize the challenges of detecting defects of varied size and shape, under uneven illumination and high background similarity. In this sense, both domains are challenging, though insulators add additional complexity due to multi-material and dynamic environmental interference.
I suggest the following revised wording, which better aligns with the paper’s contribution:
Proposed new wording:
“It is worth noting that some traditional methods have shown advantages in complex backgrounds, such as edge extraction technology based on Neutrosophic Canny segmentation, which can achieve precise localization of small to large defects in structured settings (e.g., TFT-LCD panels) through refined operator design [10]. In this study, we explored the power of convolutional neural networks (CNNs) to improve defect detection performance under diverse conditions.”
Author Response
Dear Reviewer, We would like to start by extending our most sincere gratitude for your meticulous and thoughtful feedback. Your sharp insight into the flaws in our References has been incredibly precious to refining our manuscript, and we truly appreciate the time and care you invested in pointing out these details. We offer our humble and earnest apologies for our failure to accurately present the content related to this reference in the initial version. This oversight resulted in unnecessary ambiguities, and we feel truly regretful for any confusion or extra effort this may have caused you during your review. Your reminder has made us clearly recognize a critical gap in the clarity and precision of our writing, and we take full responsibility for this shortcoming. We have carefully revised the corresponding part in the Introduction section (Page 2, Lines 70–75 of the main text) as guided by your feedback.The updated content is as follows:“It is worth noting that some traditional methods have shown advantages in complex backgrounds, such as edge extraction technology based on Neutrosophic Canny segmentation, which can achieve precise localization of small to large defects in structured settings (e.g., TFT-LCD panels) through refined operator design [10]. In this study, we explored the power of convolutional neural networks (CNNs) to improve defect detection performance under diverse conditions.” Your suggestion has been of immense benefit to our work—and this is no overstatement. It has not only directly resolved the ambiguity in the original description but also significantly enhanced the precision of our research background elaboration and the rigor of our academic expression, making the overall logic of the manuscript more coherent and credible. If you have any further questions about the revised content, or if you believe additional adjustments are necessary, please do not hesitate to inform us. We are committed to responding as promptly as possible and will spare no effort to address every concern of yours thoroughly and carefully. Once again, we would like to express our deepest thanks for your patience, understanding, and invaluable guidance. Sincerely,
shuaishuai li
This manuscript is a resubmission of an earlier submission. The following is a list of the peer review reports and author responses from that submission.
Round 1
Reviewer 1 Report
Comments and Suggestions for Authors
In this article, a feature focused diffusion network is proposed for Insulator defect detection. Except some experiment data, no defect image and result image of detection appear in the article.
Comments on the Quality of English Language- In Abstract, line13: “make each scale have rich context information, and improve the overall detection effect of the network, and improve the overall detection effect of the network” is confusing.
- Figure 1 (IDD-DETR network architecture) is not clear enough and too cluttered. It can be described more comprehensively to help readers understand the role of each component, and the spelling of "rulu" in the figure should be "ReLU"Model.
- In line 161 of P 3.1, "the ap50 of flash defects in existing methods is generally less than 70%," the term "ap50" should be changed to "mAP50," which is a more standard term in object detection and multiple identical errors appear throughout the text.
- In line 164 of P 3.1, 'the traditional two-way fusion path (top-down+bottom up) lacks adaptive adjustment to the difference of defect scale': 'defect scale' should be changed to 'feature scale'. In practical applications, the focus is usually on the size of the target or the scale of the features, rather than 'defect scale'.
- In line 193 of P 3.2.1, formula 1 does not define γ, and the formula layout is unclear.
- In line 403, P 4.3.4, 'to verify the generalization ability of the model (table 6-7)' and 'table 6-7' should be capitalized as' Table 6-7'.
Reviewer 2 Report
Comments and Suggestions for Authors
Overall evaluation: This paper addresses the practical engineering challenge of detecting defects in transmission line insulators, targeting the core issue of insufficient detection accuracy caused by significant target scale variation and complex backgrounds. The proposed improved RT-DETR framework incorporates the Feature Focused Diffusion Network (FFDN) and Dynamic Range Histogram Self-Attention (DHSA) to address problems of inadequate multi-scale feature fusion and low-contrast defect feature suppression. A specialized dataset with 15% flashover defects is developed to address small-sample detection limitations. Experimental results show that the method achieves a 4.7% increase in mAP50 and a 6.1% improvement in flashover defect detection accuracy while maintaining real-time performance at 76 FPS, enhancing the model’s adaptability to small targets and complex environments. The paper presents a clear technical approach with defined innovations and rigorous experimental validation, offering both theoretical contributions in multi-scale feature fusion and dynamic attention mechanisms and practical solutions for UAV-based inspection scenarios. While the research is structurally sound and logically consistent, minor refinements to textual clarity, terminology standardization, and literature integration are recommended to enhance academic rigor.
Specific Suggestions:
- Abstract Module Advantages:Clearly state the core benefits of FFDN and DHSA in the abstract, emphasizing their unique roles in addressing multi-scale feature fusion and low-contrast defect detection to highlight the study’s innovations.
- Introduction Practical Context:Expand the introduction to detail challenges in UAV-based insulator inspection, such as small defect scale and complex backgrounds, to reinforce the real-world urgency of the research problem.
- Literature Review Updates:Incorporate recent research on lightweight networks, multi-scale fusion, and dynamic attention mechanisms, analyzing existing methods’ limitations to justify the proposed improvements.
- Visual Notation Consistency:Ensure figures/tables use full module names alongside abbreviations at first mention, simplifying subfigure descriptions for clarity.
- Formula Symbol Annotations:Add brief definitions for new symbols in formulas when they first appear to avoid ambiguity and aid reader understanding.
- Dataset Acquisition Details:Include specific collection parameters in the dataset description to enhance reproducibility and representativeness.
- Conclusion Future Directions: Clearly outline actionable future work in the conclusion, such as technical extensions or application scenarios, to demonstrate the study’s potential for broader impact.
Overall Recommendation: The research makes a valuable contribution to insulator defect detection with its innovative framework and practical insights. The proposed modifications focus on improving textual clarity, contextual relevance, and scholarly standards, all of which are minor yet important adjustments. By addressing these points, the paper will become more precise, rigorous, and accessible to readers. Recommended for acceptance after minor revisions.
Reviewer 3 Report
Comments and Suggestions for Authors
The manuscript proposes an improved insulator defect detection network (IDD-DETR) based on RT-DETR, incorporating three new modules (FFDN, DASI, DHSA). While the topic is relevant and timely, the paper has major flaws in clarity, novelty, and scientific rigor that prevent it from being considered for publication in its current form.
-
The manuscript is difficult to read due to persistent grammatical errors, awkward phrasing, and unclear technical descriptions. Many sections require substantial rewriting for basic comprehension.
-
The proposed modules are only minor adaptations of existing designs (e.g., BiFPN, ASFF, DRFPN). The novelty is neither clearly articulated nor convincingly demonstrated.
-
Key components such as DHSA and DASI are described in a convoluted manner, with insufficient mathematical clarity or visual explanation.
-
The dataset used is small and imbalanced, and no statistical confidence or reproducibility analysis is provided. Results lack error bars or variance measures and may not be generalizable.
-
The paper compares to an excessive number of YOLO variants with unclear relevance, while fair comparisons to closely related transformer-based methods are missing or insufficient.
-
Several claims, such as “1.7× signal-to-noise improvement” or “significant robustness”, are made without rigorous supporting evidence. Performance improvements are modest and not statistically validated.
Reviewer 4 Report
Comments and Suggestions for Authors
The following issues should be addressed by the authors:
-It will be helpful to explain why FFDN and DHSA perform better than current models regarding accuracy and efficiency.
-Emphasise real-world deployments of FFDN and DHSA on edge devices towards environmental robustness and inference latency.
-Baseline models (e.g., CNNs, Vision Transformers), FFDN, and DHSA can be contrasted in terms of accuracy (e.g., top-1 accuracy, mAP), and computational complexity.
-In terms of practicalities like durability and battery life, report in terms of model deployment on drones and edge devices across different environmental conditions.